# Knockdown resistance allele L1014F introduced by CRISPR/Cas9 is not associated with altered vector competence of *Anopheles gambiae* for o'nyong nyong virus

**Grant A. Kay**[1], **Edward I. Patterson**[2], **Grant L. Hughes**[1], **Jennifer S. Lord**[1], **Lisa J. Reimer**[1]*

1 Department of Vector Biology, Liverpool School of Tropical Medicine, Liverpool, United Kingdom,
2 Biological Sciences, Brock University, St. Catharines, Ontario, Canada

* lisa.reimer@lstmed.ac.uk

**Data Availability Statement:** Data are available in Open Science Framework: https://doi.org/10.17605/OSF.IO/3NFR7.

## Abstract

Knockdown resistance (*kdr*) alleles conferring resistance to pyrethroid insecticides are widespread amongst vector populations. Previous research has suggested that these alleles are associated with changes in the vector competence of mosquitoes for arboviruses and *Plasmodium*, however non-target genetic differences between mosquito strains may have had a confounding effect. Here, to minimise genetic differences, the laboratory *Anopheles gambiae* Kisumu strain was compared to a CRISPR/Cas9 homozygous *kdr* L1014F mutant Kisumu-*kdr* line in order to examine associations with vector competence for o'nyong nyong virus (ONNV). Mosquitoes were infected using either blood feeds or intrathoracic microinjections. There were no significant differences in the prevalence of virus in mosquito body parts between *kdr* mutant and wildtype lines from either oral or intrathoracic injection routes. The ONNV titre was significantly higher in the legs of the wildtype strain at 7dpi following intrathoracic microinjection, but no other significant differences in viral titre were detected. ONNV was not detected in the saliva of mosquitoes from either strain. Our findings from per os infections suggest that the *kdr* L1014F allele is not associated with altered infection prevalence for ONNV, a key component of vector competence.

## Introduction

Pyrethroid insecticides are widely used to target mosquito species that transmit human pathogens such as arboviruses and *Plasmodium*. They exert insecticidal effects through binding to receptor sites on voltage-gated sodium channels (VGSC) of mosquito neurones, resulting in prolonged channel opening, repeated nerve discharges and disrupted neurological functioning [1]. Mutations to the pyrethroid binding sites of the VGSC can confer reduced pyrethroid binding, which may result in reduced phenotypic sensitivity to pyrethroids known as knockdown resistance (*kdr*) [1]. To date, several amino acid substitutions have been associated with knockdown resistance in *Anopheles* species, including substitutions of leucine at codon position 1014 for phenylalanine (L1014F) or serine (L1014S) [1]. These mutations, and many

**Funding:** GK was supported by the Medical Research Council Doctoral Training Programme at LSTM (grant number N013514/). The funders had no role in study design, data collection and analysis, decision to publish, or preparation of the manuscript.

**Competing interests:** The authors have declared that no competing interests exist.

others, are now commonly found across sub-Saharan Africa [2]. In addition to their ability to confer insecticide resistance, concerns have been raised that these mutations may influence the vector competence of mosquitoes for human pathogens. Vector competence describes the ability of a vector to acquire, incubate, and transmit a pathogen, and is a complex summation of many interlinked determinants including vector and pathogen genetics, microbiome, and environmental factors [3–5].

Given that *kdr* are simple amino acid substitutions in the VGSC of mosquito neurones, it is not immediately apparent how this alteration could feasibly influence vector competence. Several putative mechanisms have been suggested, ranging from pleiotropic effects of *kdr*, to indirect effects relating to broader changes in physiology. It has been speculated that a mutated VGSC may be directly capable of altering the immune response of vectors to pathogens through modifying the neurohormonal control of immune function. For example, the gene expression of 20-hydroxyecdysone is under neurohormonal control and has been shown to increase immune gene transcription and alter *Plasmodium berghei* infection rate in *Anopheles gambiae* [6–8]. Less direct mechanisms have also been proposed. The presence of *kdr* has been linked to multiple fitness costs in mosquito vectors including development times, longevity, reproduction success and feeding behaviours [9–12]. Given that vector competence is a complex balance of multiple known and unknown determinants, it is conceivable that mutated VGSC could indirectly influence vector competence through such physiological disturbances. A further possibility is that the evolutionary conditions favouring the emergence and spread of *kdr*, may also co-select for alleles on local or distant loci that can impact vector competence [10, 13]. For example, the *kdr* mutation occurs in a haplotype that contains a gene coding for the serine protease *ClipC9*, which has been linked with anti-*Plasmodium* immunity [14, 15]. It has also been shown in genome wide association studies with *Aedes aegypti* that there are single nucleotide polymorphisms commonly associated with *kdr* in distant loci of the genome, which could potentially influence vector competence [16].

Several studies have investigated the potential impacts of *kdr* on vector competence to date, but findings have been varied and conflicting, likely due to the range of vector-parasite models used, and significant differences in study design. Whilst there are relatively few studies that have focused on the potential impacts of *kdr* on arbovirus transmission, inferences may also be drawn from several field and laboratory investigations conducted using *Plasmodium* parasites and their mosquito vectors. The presence of the L1014S allele was associated with a higher prevalence of *P. falciparum* sporozoites in field-caught *An. gambiae s.s.* from Tanzania [17] but similar studies have not detected any significant effect [18, 19]. Infection prevalence data from field caught mosquitoes are vulnerable to the confounding effects of insecticide use in the study area. *Kdr*-carrying mosquitoes may be more likely to survive the extrinsic incubation period (EIP) of a parasite in areas where insecticide use is common and may therefore be more likely to harbour malaria sporozoites than those with wildtype VGSC. Conversely, *Plasmodium*-infected mosquitoes with *kdr* alleles may be more vulnerable to the effects of pyrethroids that their uninfected counterparts, making survival through a parasite's EIP less likely in areas of insecticide use [20].

Experimental infection of field mosquitoes in a laboratory can address some of these issues. Ndiath et al. [21] reported that *An. gambiae s.l.* carrying the L1014F or L1014S alleles, had significantly higher prevalence and density of oocysts and sporozoites than wildtype controls, but this effect was not seen in a similar study with a field-derived *Anopheles* line [22]. Experimentally infected field strains of *Ae. aegypti* from Florida with differing *kdr* genotypes (L1016 and C1534) had different infection and transmission rates for dengue virus, with the strains with the highest proportion of *kdr* demonstrating the lowest vector competence indices [23].

Similar results were observed in isofemale lines derived from field populations, whereby strains with the highest prevalence of 1534C alleles had the lowest virus dissemination rate at 7dpi, and viral loads were not significantly different to the wildtype strains [24]. However, the main limitation of all studies using field-caught mosquitoes is that despite the thorough characterisation of strains by *kdr* genotype, they are not able to account for any potential effects of other genetic differences that could alter vector competence.

This problem is partially addressed by developing laboratory strains of mosquitoes with differing *kdr* profiles. Chen et al. [25] developed a permethrin resistant mosquito line from a parental field-caught *Ae. aegypti* strain using insecticide selection. The resistant strain had both a significantly higher prevalence of mutated F1534C and V1016I, and a higher dissemination rate of dengue virus to legs at 14 days post-infection (dpi) than the unselected strain [25]. However, the use of insecticide selection here may also have led to unintended genetic differences between resistant and susceptible strains which could affect vector competence [4]. As such, introgression of *kdr* alleles into susceptible mosquito populations has been used to investigate the impacts on vector competence. L1014F was introgressed into the wildtype *An. gambiae s.s.* Kisumu strain, and *kdr* genotype was associated with increased infection prevalence at both the oocyst and sporozoite stages compared to the wildtype Kisumu strain [7]. One of the few arbovirus-related studies utilised backcrossing of insecticide resistant and susceptible laboratory mosquito strains to investigate the impacts of V1016I and F1534C alleles on the competence of *Ae. aegypti* for Zika virus [26]. The results show that *Ae. aegypti* with these alleles had greater infection rates and higher viral dissemination to the legs than wildtype mosquitoes at multiple timepoints post-infection [26]. Whilst introgression is a more targeted way of introducing alleles of interest into wildtype lines compared to insecticide selection, it is possible that confounding alleles are also inadvertently introgressed in the process.

In this study, to minimise confounding genetic differences, we compared vector competence of an L1014F homozygotic line of *An. gambiae* produced using CRISPR/Cas9 to its parent VGSC wildtype strain. Both strains were experimentally infected with o'nyong nyong virus (ONNV) via infected blood meals or intrathoracic injections. ONNV is the only known pathogenic arbovirus vectored by *Anopheles* spp. in sub-Saharan Africa and has a broadly overlapping distribution with mosquito populations carrying *kdr* [2, 27].

## Methods

### Virus and mosquito strains

ONNV UgMp30 (BEI Resources) was passaged once in Vero CCL81 cells for the intrathoracic injection experiments. For the oral infections, ONNV UgMp30 was passaged twice in Vero cells. Two mosquito lines were used in this study: the insecticide susceptible laboratory Kisumu strain of *An. gambiae* (hereafter Kis) with non-mutated VGSC; and a CRISPR/Cas9 gene-edited L1014F *kdr* homozygous Kisumu line (hereafter Kis-kdr). The Kis-kdr line was developed by colleagues at the Liverpool School of Tropical Medicine, and the methods are described in detail elsewhere [28]. Briefly, Kisumu eggs were injected with donor plasmids containing a red fluorescent protein (RFP) marker. RFP-expressing $G_0$ larvae were backcrossed with Kisumu to obtain a $G_1$ line that was screened for the L1014F mutation. Progeny of $G_1$ females positive for L1014F by locked nucleic acid (LNA) polymerase chain reaction (PCR) were backcrossed with Kisumu to obtain greater numbers of individuals with the *kdr* allele. Several generations of intercrossing of heterozygous individuals achieved a 100% homozygotic line for L1014F [28].

## Insecticide susceptibility testing

Both strains of mosquitoes (Kis n = 40, Kis-kdr n = 60) were exposed at 3–5 days old to the discriminating dose of permethrin (0.75%) in WHO tube assays for 1 hour [29]. They were transferred back into holding tubes with access to 10% sucrose, and knockdown was recorded. At 24 hours post-exposure, mortality rates were determined. Controls of both mosquito strains (n = 20) were exposed to insecticide negative papers in WHO tube assays.

## Per os (PO) infections

Mosquitoes aged 5–7 days were housed in cardboard soup cups and sugar starved for 24 hours with access to water. ONNV stocks were diluted 1:1 with human blood to achieve a final viral titre of $1x10^6$ pfu/ml. The phagostimulant ATP was added at a concentration of 900μM. Under Arthropod Containment Level 2 conditions, mosquitoes were allowed to blood feed via the Hemotek feeding system (Hemotek Ltd) for 1 hour. Unfed and partially fed mosquitoes were discarded, and fully fed mosquitoes were incubated until processing with access to 10% sucrose ((27˚C (+/- 1), 80% relative humidity (+/- 5), and 12:12 hours light:dark).

## Intrathoracic (IT) microinjections

Non-blood fed females aged 4–6 days were briefly cold anaesthetised at -20˚C for 30 seconds and transferred to a cold plate. Each mosquito was intrathoracically inoculated with 100nL ONNV ($1x10^5$ pfu/ml) at a rate of 50nl/second using a Nanoject III (Drummond Scientific). Following IT injection, mosquitoes were transferred to cardboard soup cups with access to 10% sucrose and incubated until further processing, as above.

## Dissections and forced salivations

For PO infections, mosquitoes were sampled at 7dpi, and heads, bodies (thorax & abdomen), and saliva were collected. To control for differences in blood meal size, whole mosquitoes from each strain were sampled immediately post-feeding to determine the viral titre of the ingested blood meal. For IT injections, a cohort of mosquitoes were sacrificed at 5, 7, or 10dpi, and head, body (thorax + abdomen), legs, and saliva were collected. For forced salivation, mosquitoes were cold anaesthetised, legs and wings removed, and the proboscis placed in a 20μL pipette tip containing mineral oil. Mosquitoes were allowed to salivate for 15 minutes, then dissected into head and bodies. Saliva was ejected into a tube containing 100 μL infection media (Dulbecco's Modified Eagle Media (DMEM) + 2% foetal bovine serum (FBS), 1:1000 v/v 50mg/mL gentamicin, 1:200 v/v 10mg/mL Fungin (Invivogen)). Body parts were placed into a Safelock tube containing 300μL infection media and a 5/32" diameter stainless steel ball bearing (Dejay Distribution Limited). Samples were immediately stored at -80˚C until further processing.

## Plaque assays

For IT injection samples, 24-well tissue culture plates were seeded with 500μL of $2.5x10^5$ cells/ml Vero CCL81 cells in growth media (DMEM + 10% FBS + 1:1000 v/v 50mg/mL gentamicin) and incubated overnight (37˚C, 5% $CO_2$). For PO infection samples, Vero cells were used, and seeded as above. Body part samples were defrosted and homogenised at 26Hz for 5 mins using a TissueLyser (Qiagen). Homogenised samples were centrifuged for 5 mins to pellet debris. Each sample was serially diluted in infection media, and 100uL inoculum was added in duplicate to the confluent cell monolayer in the cell culture plates. These were incubated for 1hr (37˚C, 5% $CO_2$) prior to adding a 0.4% agarose overlay. After 48 hours of incubation (37˚C,

5% $CO_2$), the plates were fixed with formaldehyde and stained using a 0.25% v/v crystal violet solution. Plaques were counted manually, and the mean of both replicates was used to calculate the viral titre of each sample (pfu/sample).

## Cytopathic effects assays

Cytopathic effects (CPE) assays were performed on saliva samples to determine whether ONNV was present. 96-well cell culture plates were seeded with 100μL Vero CCL81 at a density of $2.5x10^5$ cells/mL in growth media and incubated overnight. After incubation the growth media was removed and replaced with 75μL infection media. Saliva samples were vortexed for 15s and 25μL was added in duplicate to the cells. The plates were incubated for 72 hours (37˚C, 5% $CO_2$) before being scored by microscopy for the presence of CPE. If either of the replicates for each sample were positive, the saliva sample was considered to contain viable ONNV. The limit of detection of the CPE assays was established at approximately $1.9x10^2$ pfu/ml, which is <5 pfu per 25μL added to each CPE assay replicate (S1 File).

## Statistical analysis

Fishers' exact tests were used to test for significant differences in the 1 hour knockdown and 24 hour mortality of mosquito strains following exposure to 0.75% permethrin. For mosquitoes infected via the PO route, the following components of vector competence were determined. Infection prevalence was defined as the proportion of mosquitoes with established infection of their bodies (thorax and abdomen) out of the total number taking an infected blood meal. Dissemination prevalence was defined as the proportion of mosquitoes with established infection in their bodies, that developed infection in the heads or legs. Transmission prevalence was defined as the proportion of mosquitoes with disseminated infection that also had virus present in saliva. Fishers' exact tests were used to determine the significance of differences in infection and dissemination prevalence. This was justified due to the small sample size for each timepoint.

For mosquitoes infected via the IT route, the proportions of mosquitoes developing an established infection in the body (thorax & abdomen), heads, and legs, out of the total number injected, were determined. The transmission prevalence was defined as the proportion of mosquitoes developing salivary infection following IT injection with ONNV. Fishers' exact tests were used to test for significant differences in these proportions between mosquito strains.

For viral titre data of mosquito samples and ingested blood meals, normality was determined using Shapiro-Wilks test and separate ANOVA were performed for each timepoint, using viral titre and mosquito strain as grouping variables. If the viral titre data were non-Normally distributed, Kruskal-Wallis tests were used. All statistical analyses were performed using base functions in R (version 4.2.1).

## Results

### Permethrin susceptibility testing

Exposure to 0.75% permethrin in WHO tube assays revealed a significant difference in susceptibility between strains. There was 95% ([95% CI 83–99] n = 40) knockdown following 1hr exposure of Kis, compared to 3% ([0–12] n = 60) of Kis-kdr (Fishers' exact p≤2.2e-16). Mortality at 24hrs revealed a similar picture with Kis showing high susceptibility (100% mortality [91–100] n = 40), and Kis-kdr showing a significantly lower mortality (15% [7–27] n = 60) (Fishers' exact p≤2.2e16) (Fig 1). There was no knockdown or mortality seen in the unexposed control groups for either strain.

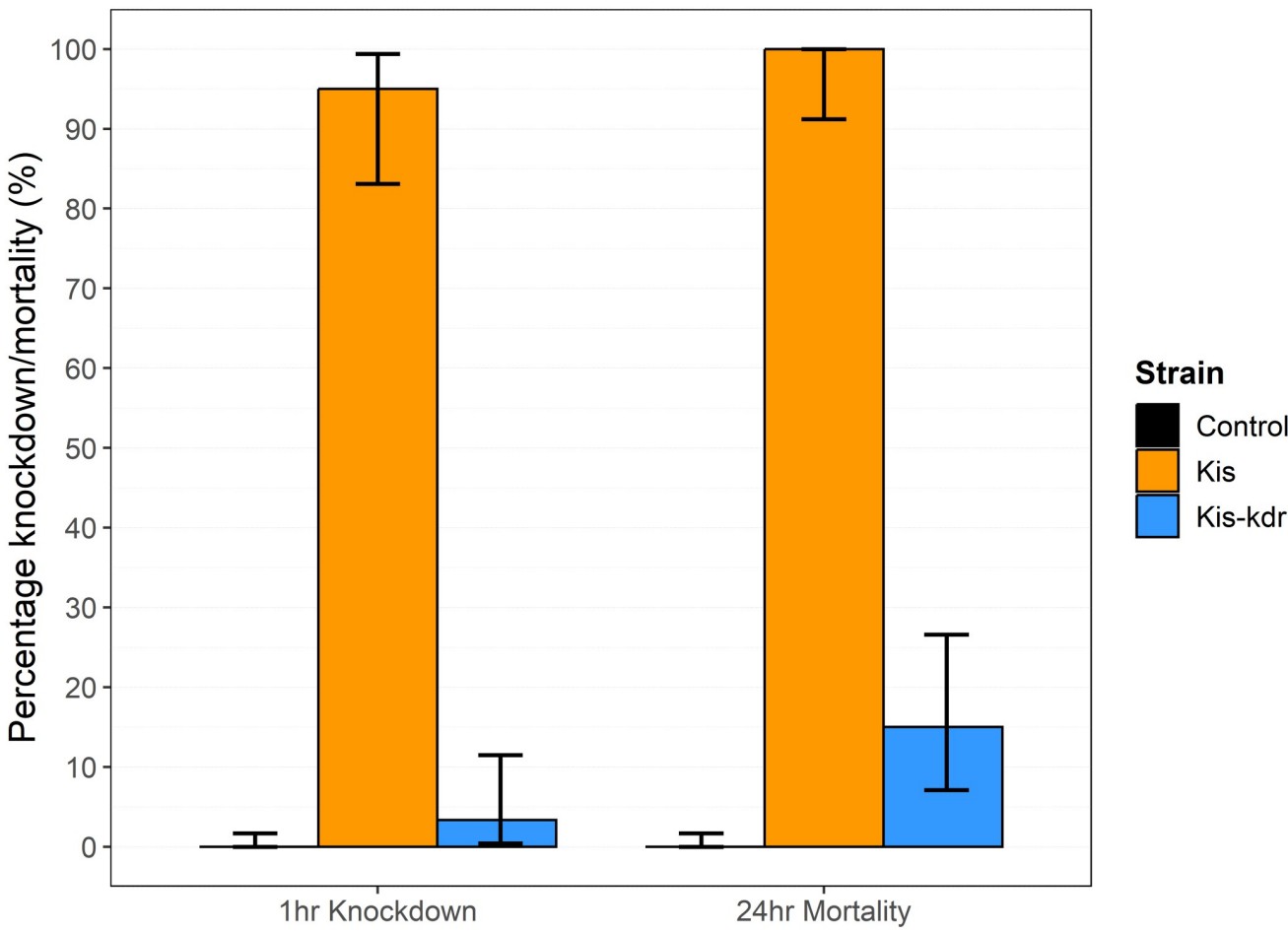

**Fig 1. Susceptibility to 0.75% permethrin in WHO tube assay.** Kis (n = 40) and Kis-kdr (n = 60) were exposed in WHO tube assays to 0.75% permethrin for 1 hour. Controls were mock-exposed to non-insecticide treated filter papers. Percentage knockdown at 1hr, and mortality at 24hr are indicated by the coloured bars. 95% confidence intervals are marked in black. There was significantly higher (Fisher's exact p≤2.2e-16) knockdown at 1 hour in Kis (95% [95% CI 83–99]) than Kis-kdr 3% [0–12]). Mortality at 24 hours was significantly higher in Kis (100% [91–100] than Kis-kdr (15% [7–27]) (Fisher's exact p≤2.2e-16). There was no knockdown or mortality in negative controls of either strain.

## Oral infections

Homogenisation of mosquitoes immediately following blood feeding revealed a similar range of inoculum titres, spanning from approximately 3.2 to $4.2\log_{10}$ pfu/mosquito, in both strains. The mean inoculum titres were not significantly different between strains (Kis 3.77 [95% CI 3.59–3.95] n = 15; Kis-kdr 3.67 [3.52–3.82] n = 23; ANOVA p = 0.40).

Establishment of infection via infected blood meal was achieved in both mosquito strains. The infection prevalence was not significantly different between Kis (45% [23–69] n = 20) and Kis-kdr (40% [19–64] n = 20) (Fishers' exact p = 1). At the 7dpi timepoint, there was very limited dissemination of virus to other body parts in both mosquito strains, with only a single head sample testing positive by plaque assay, and no positive leg samples. These differences were not significantly different between strains (Fishers' exact p = 1). Of the mosquitoes with established infection at 7dpi, the mean body ONNV titre was similar in both strains (Kis-kdr (2.30 $\log_{10}$ pfu/body), Kis (1.99); ANOVA (p = 0.57) (Fig 2). The only mosquito with a disseminated infection to the head was of the Kis strain and had a body ONNV titre of 2.41 ($\log_{10}$

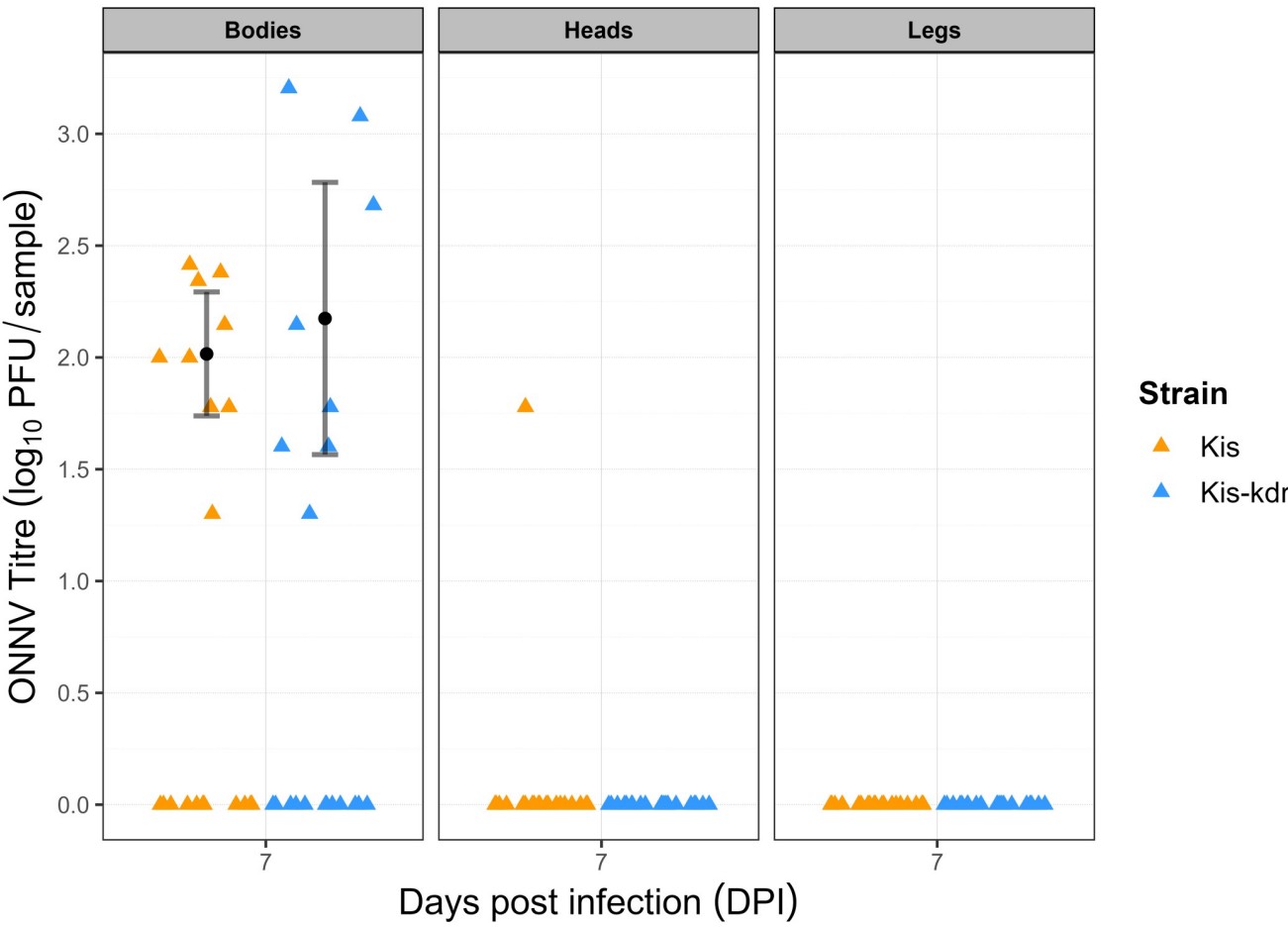

**Fig 2. ONNV titre following PO infection.** Viral titres of mosquito bodies, heads, and legs at 7dpi were established by plaque assay for Kis (n = 20) and Kis-kdr (n = 20) strains. The mean body ONNV titres were not significantly different between strains (Kis 1.99 $\log_{10}$ pfu/sample; Kis-kdr 2.30; ANOVA p = 0.57). Only one Kis mosquito had a disseminated infection to its head at 7dpi. No virus was found in the legs of either mosquito strain at 7dpi. The mean values and 95% confidence intervals are marked in black.

pfu/body) which was the highest body titre of any Kis mosquito assayed, but the saliva sample was negative for virus. Interestingly there were three Kis-kdr mosquito bodies which had higher titres but did not exhibit disseminated infections to their heads by 7dpi. Unfortunately, there were insufficient numbers of mosquitoes surviving beyond 7dpi to allow for further timepoints to be conducted.

### Intrathoracic injections

For the intrathoracic injection experiments, a sample size of >20 mosquitoes per strain was achieved for all timepoints except Kis-kdr at 10dpi. ONNV was detected in 100% of mosquito bodies and heads, at all timepoints for both mosquito strains. The majority of leg samples were ONNV positive in both strains at all timepoints (5dpi Kis 91% [95% CI 71–99], Kis-kdr 96% [77–100]; 7dpi Kis 100%, Kis-kdr 100%; 10dpi Kis 96% [78–100], Kis-kdr 93% [66–100]). None of these differences in between mosquito strains were statistically significant (Fishers' exact p = 1).

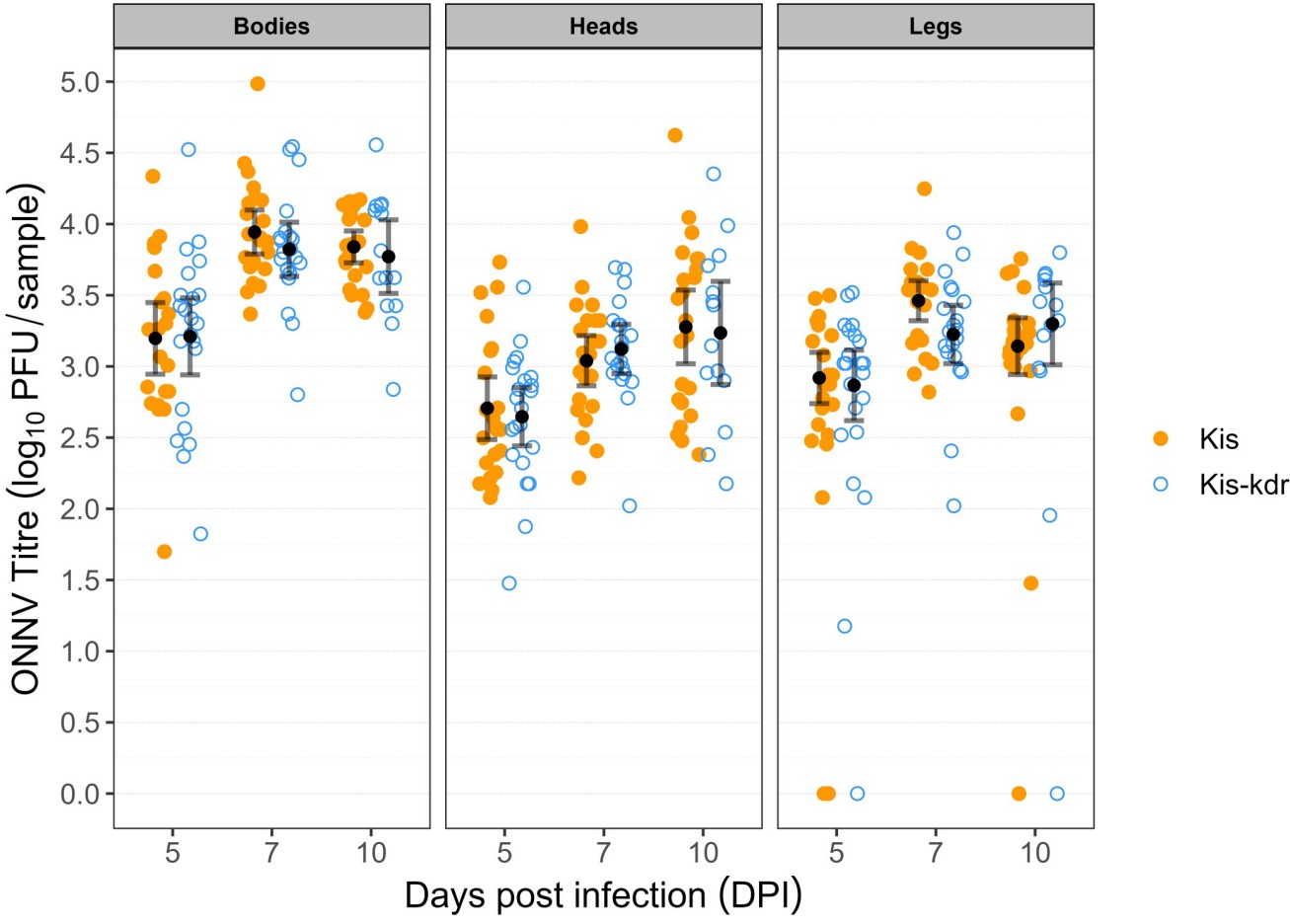

**Fig 3. ONNV titre in mosquito body parts following intrathoracic microinjection.** Mean viral titres and 95% CI shown in black (virus negative samples excluded from calculations). The inoculum delivered by intrathoracic microinjection was 100nL of $10^5$ pfu/mL.

The mean viral loads in each tissue at each timepoint followed the same broad pattern for each mosquito strain (Fig 3). Following inoculation of approximately 10 pfu (100nL of $10^5$ pfu/mL) of ONNV into the haemocoele, there was rapid amplification of virus in the body parts of mosquitoes. There was an observed plateauing of viral titre in bodies and legs by the 10dpi timepoint. The viral titre in the heads of mosquitoes continued to increase up to 10dpi. There was a significant difference in the viral titre of the legs of mosquitoes at 7dpi (Kis mean titre 3.46 $\log_{10}$ pfu/sample [3.32–3.60]; Kis-kdr 3.22 [3.01–3.42]; ANOVA p = 0.049) (Fig 3). This significant difference was no longer observed at 10dpi. There were no significant differences in the viral titres in other body parts at any timepoint. CPE analysis of saliva samples revealed no positive samples at any timepoint in either strain.

## Discussion

Mutations to voltage-gated sodium channels, associated with resistance to pyrethroid insecticides, have previously been correlated with alterations in vector competence for *Plasmodium* spp. and some arboviruses [7, 17, 21, 23–26]. Here we investigated the potential impacts of the L1014F allele on the vector competence of *An. gambiae* for ONNV. Susceptibility testing to

permethrin demonstrated a significant phenotypic difference between wildtype and *kdr*-mutant lines, which corroborates previous evaluation of these mosquito lines, and implicates the L1014F in causing permethrin resistance [28].

Our data show that there was no impact of the L1014F allele on the ONNV infection prevalence in the bodies of mosquitoes following PO infection. Parker-Crockett et al. [26] reported a higher infection prevalence for ZIKV in *Ae. aegypti* carrying the 1016 and 1534 *kdr* alleles following introgression. Our data suggest that this is not the case with the L1014F allele in this *Anopheles*-ONNV model. Previous investigations with laboratory and field mosquito strains have reported that the L1014F allele does not lead to alterations in vector competence [18, 19, 22, 30]. Our results may support this conclusion, but due to a lack of dissemination of ONNV from the midgut following PO feeds in both Kis and Kis-kdr strains, we cannot exclude any potential effects of the L1014F allele on transmission. The almost universal lack of dissemination of virus from the midgut of mosquitoes following oral infection, irrespective of their *kdr* genotype, suggests that the Kisumu strain is not a highly competent vector for this strain of ONNV. It is possible that further incubation time beyond the 7dpi tested may be required for dissemination from the midgut to occur, or dissemination may be inhibited by the presence of a midgut escape barrier [31].

Data from IT injections showed that both strains were able to support viral replication in the haemocoele, and other body parts, when the midgut was bypassed. Excluding a transiently lower mean ONNV titre in the legs of the KDR strain at 7dpi following IT injection, viral titres were not significantly different between strains. We did not detect viable ONNV in saliva samples from either mosquito strain at any timepoint post-IT injection. Alphaviruses, including ONNV, can be difficult to isolate via forced salivation, presumably due to lower salivary viral loads than with other genera of arboviruses [32–35]. It is not clear here whether additional incubation time post-infection was required for virus to enter the saliva, or whether this finding represents the presence of salivary gland infection and/or escape barriers which could have excluded ONNV from the saliva [36].

In contrast to our findings, a number of investigations have reported that *kdr* is associated with altered vector competence indices [7, 17, 21, 23–26]. There are several reasons for these discordant findings. Genetic differences were minimised between *kdr*-mutant and wildtype strains through the use of a gene edited *kdr*-mutant line in our study. Though the CRISPR/Cas9 technique is not completely immune to the introduction of unintended genetic changes through off-target activity and genetic drift [37, 38], it limits the introduction of confounding alleles compared to backcrossing, insecticide selection, or comparisons between sympatric field strains. The effects observed in previous investigations employing these techniques may have been due to other genetic differences rather than *kdr* genotype.

In summary, our findings from PO infections suggest that the *kdr* 1014F allele, introduced by CRISPR/Cas9, is not associated with alterations to ONNV infection prevalence, a key component of vector competence. Further investigations should isolate the potential effects of *kdr* by aiming to minimise potentially confounding genetic differences between mosquito strains. Whilst aspects of vector competence for ONNV may not be significantly modified by the introduction of the L1014F allele in these *An. gambiae* strains, this does not preclude potential alterations to broader aspects of vectorial capacity. Previous evaluation of the Kis-kdr strain noted a number of fitness costs associated with introduction of the L1014F allele in the absence of insecticide pressures, including a significantly reduced average lifespan [28]. This could reduce the probability of a mosquito successfully surviving the EIP and being capable of transmitting a pathogen. Conversely, in the presence of insecticide, *kdr*-carrying mosquitoes may be more likely to survive the duration of the EIP and could have a higher resulting vectorial capacity than non-mutant populations.

## Supporting information

**S1 File. Cytopathic effects assay sensitivity analysis.**
(DOCX)

## Acknowledgments

We would like to thank Dr Linda Grigoraki of the Liverpool School of Tropical Medicine (LSTM) for providing the mosquito strains used in this experiment. We would also like to thank Ruth Cowlishaw and Sara Elg (both of LSTM) for their help and guidance with mosquito rearing.

## Author Contributions

**Conceptualization:** Grant A. Kay, Lisa J. Reimer.

**Formal analysis:** Grant A. Kay, Jennifer S. Lord.

**Investigation:** Grant A. Kay.

**Methodology:** Edward I. Patterson.

**Resources:** Grant L. Hughes, Lisa J. Reimer.

**Supervision:** Lisa J. Reimer.

**Writing – original draft:** Grant A. Kay.

**Writing – review & editing:** Grant L. Hughes, Jennifer S. Lord, Lisa J. Reimer.

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
