## [Decision Letter · Decision Letter 0]

12 Mar 2023

PONE-D-23-03395Knockdown resistance allele L1014F introduced by CRISPR/Cas9 is not associated with altered vector competence for O’nyong nyong virus in Anopheles gambiae”PLOS ONE

Dear Dr. Reimer,

Thank you for submitting your manuscript to PLOS ONE. After careful consideration, we feel that it has merit but does not fully meet PLOS ONE’s publication criteria as it currently stands. Therefore, we invite you to submit a revised version of the manuscript that addresses the points raised during the review process.

The major weakness of this paper is that the original mosquito strain seems not to be sensitive to the  ONNV that largely question the usefulness of the data, however, thanks to the carefull review that was done by the reviewer, the results seems to deserve to be published we moderation of the conclusion. This is an impact on infection susceptibility rather than on competence efficiency as highlited.

We look forward to receiving your revised manuscript.

Kind regards,

Pierre Roques, Ph.D.

Academic Editor

PLOS ONE

Journal Requirements:

Reviewers' comments:

Reviewer's Responses to Questions

**Comments to the Author**

1. Is the manuscript technically sound, and do the data support the conclusions?

Reviewer #1: Partly

2. Has the statistical analysis been performed appropriately and rigorously? 

Reviewer #1: Yes

3. Have the authors made all data underlying the findings in their manuscript fully available?

Reviewer #1: Yes

4. Is the manuscript presented in an intelligible fashion and written in standard English?

Reviewer #1: Yes

5. Review Comments to the Author

Reviewer #1: The article presented here is well introduced and the objective is clear. The proposed method is original and well justified (use of CRISPR/Cas9 to introduce a SNP into a line, allowing to have two lines with close genetic backgrounds varying only by this SNP). I cannot judge the method of creating the strain by CRISPER/Cas9. I am not a specialist in this field. The creation of the strain having been the subject of another published article, I consider that it has already been evaluated and validated by other reviewers.

Several points should be clarified in the method section, and I propose several modifications to the analysis and presentation of the results, notably not to compare your 7 dpi (reflecting the replication of the virus in the mosquito) with the "0 dpi" (control of the titre of the blood meal provided to the mosquitoes).

From a general point of view, I think that the title and the conclusions should be slightly moderated by the authors. Indeed, the Kis strain (without kdr) infects well (45% of bodies positive at 7 dpi) but does not disseminate and no saliva is positive. This strain is therefore not competent for the ONNV strain used. The same results were observed for the Kis-kdr strain. But can we really speak of an absence of impact of kdr on vector competence, given that the Kis strain is already not competent?

I would therefore say that “kdr has no impact on infection” rather than “on competence”.

Also, be careful to not over-interpret IT results. I agree that it is an interesting addition method, but IT does not measure vector competence (i.e., no measurement of EIP).

specific comments:

Line 32: put kdr in italics.

Line 42: I would delete "modest". It is indeed a SNP, but kdr has a major impact on the survival of Anopheles under insecticide pressure.

Line 58 - 60: Although the cited paper does not determine the role of these kdr-associated SNPs, perhaps specify that they could be associated with a reduced cost on kdr fitness? At present, it is not clear what information this sentence provides.

Line 130-134: the numbers are a bit limited for this type of test but seem sufficient from figure 1.

Line 152: at 0 dpi? specify that this is to check the titre of the inoculum.

Line 171: please describe a little better what is counted here and how the viral titre is calculated or cite a reference describing the method.

Line 190: cite R rather than R studio (type > citation()" in the console).

Line 193: Throughout the paragraph, you can round percentages without decimal (the numbers are small, so such precision is unnecessary). Do not write XX.0%.

Line 214-215: Please round the percentages. The numbers are also very small (n=20).

Line 218: Was the Fisher test done on a proportion calculated on n=20 per class, or only on the infected (n=8-9)? This is not specified in the text.

Line 221-222: I do not understand this comparison. Line 209-212 states that the 0 dpi test was done on whole mosquitoes to check the viral titre of the blood meal. This is a experimental control. It is not possible to compare a whole mosquito assay that has just completed a blood meal (mosquito + blood + virus) with the mosquito body at 7 dpi (reflecting mosquito infection and viral replication). This sentence should be deleted, and Figure 2 should be amended by deleting the 0 dpi and the whole mosquito category.

Lines 229-228: to be amended (see previous comment).

Line 240-246: same comment on percentages to be rounded.

Line 246-247: I think the sentence is incomplete.

Line 248-256: If the statistical tests are not significant, it means that the H0 hypothesis (no difference) cannot be rejected in view of your results. From there, why talk about an increase in viral titre? This section should be moderated in my opinion.

Figure 3: As for figure 2, representing the 0 dpi does not make sense. It is not a variable measured in the experiment, but an initial condition of the manipulation. In my opinion, 0 dpi should be removed from the figure.

Line 297 - 298: ITs are not a measure of vector competence (i.e., EIP), but only of the susceptibility of mosquitoes to infection.

Line 309 - 310: Yes, I agree with you on this point.

6. PLOS authors have the option to publish the peer review history of their article (what does this mean?). If published, this will include your full peer review and any attached files.

Reviewer #1: No

---

## [Author Response · Author response to Decision Letter 0]

28 Apr 2023

(our response was also uploaded as a word document)

We thank the reviewer for their helpful and thorough comments. We have indicated below where changes have been made, with line numbers referring to the version with tracked changes.

Overall comments

The article presented here is well introduced and the objective is clear. The proposed method is original and well justified (use of CRISPR/Cas9 to introduce a SNP into a line, allowing to have two lines with close genetic backgrounds varying only by this SNP). I cannot judge the method of creating the strain by CRISPER/Cas9. I am not a specialist in this field. The creation of the strain having been the subject of another published article, I consider that it has already been evaluated and validated by other reviewers. Several points should be clarified in the method section, and I propose several modifications to the analysis and presentation of the results, notably not to compare your 7 dpi (reflecting the replication of the virus in the mosquito) with the "0 dpi" (control of the titre of the blood meal provided to the mosquitoes).

Comparisons with 0 dpi have been removed from the figure, Methods (line 155) and Results (line 246). These data are now only presented as part of our internal controls.

From a general point of view, I think that the title and the conclusions should be slightly moderated by the authors. Indeed, the Kis strain (without kdr) infects well (45% of bodies positive at 7 dpi) but does not disseminate and no saliva is positive. This strain is therefore not competent for the ONNV strain used. The same results were observed for the Kis-kdr strain. But can we really speak of an absence of impact of kdr on vector competence, given that the Kis strain is already not competent? I would therefore say that “kdr has no impact on infection” rather than “on competence”.

We have moderated the text as suggested, clarifying that the measured effects are on infection rather than competence. For example in the abstract we have updated our conclusion to “Our findings from per os infections suggest that the kdr L1014F allele is not associated with altered infection prevalence for ONNV, a key component of vector competence” (Line 24). We have also defined all components of competence, including infection, dissemination and transmission (from line 195).

Also, be careful to not over-interpret IT results. I agree that it is an interesting addition method, but IT does not measure vector competence (i.e., no measurement of EIP).

We agree with the reviewer that IT cannot quantify the EIP that would occur during oral infection due to bypassing the midgut barrier. We have now clearly defined the utility of IT injections in the study design, which allows us to measure virus replication and the salivary gland infection barrier. We have separated the analysis from per os infections and clarified in the discussion what each experiment contributes to the study conclusions. 

specific comments 

Line 32: put kdr in italics.

- This has been amended

Line 42: I would delete "modest". It is indeed a SNP, but kdr has a major impact on the survival of Anopheles under insecticide pressure.

- The word modest has been deleted 

Line 58 - 60: Although the cited paper does not determine the role of these kdr-associated SNPs, perhaps specify that they could be associated with a reduced cost on kdr fitness? At present, it is not clear what information this sentence provides.

- Further explanation for the inclusion of this citation has been added to line 61. 

Line 130-134: the numbers are a bit limited for this type of test but seem sufficient from figure 1.

- The insecticide resistance phenotypes of these strains have been fully characterised elsewhere (reference 28). We feel that the marked difference in susceptibility to 0.75% permethrin exposure is sufficient to demonstrate phenotype.

Line 152: at 0 dpi? specify that this is to check the titre of the inoculum.

- 0dpi has been removed, clarifications have been added that the screening at 0dpi was to determine viral titre of the inoculum.

Line 171: please describe a little better what is counted here and how the viral titre is calculated or cite a reference describing the method.

- We followed standard virological methods and have now clarified that the plaques were manually counted and results are presented in mean number of plaque forming units per sample (line 174). 

Line 190: cite R rather than R studio (type > citation()" in the console).

- This has been updated

Line 193: Throughout the paragraph, you can round percentages without decimal (the numbers are small, so such precision is unnecessary). Do not write XX.0%.

- Percentages have been rounded 

Line 214-215: Please round the percentages. The numbers are also very small (n=20).

- Amended

Line 218: Was the Fisher test done on a proportion calculated on n=20 per class, or only on the infected (n=8-9)? This is not specified in the text.

- Fisher’s test was performed on the proportion infected out of the total number successfully blood feeding (n=20)

- For clarity, definitions of infection, dissemination and transmission have been added to the methods section (from line 190)

Line 221-222: I do not understand this comparison. Line 209-212 states that the 0 dpi test was done on whole mosquitoes to check the viral titre of the blood meal. This is a experimental control. It is not possible to compare a whole mosquito assay that has just completed a blood meal (mosquito + blood + virus) with the mosquito body at 7 dpi (reflecting mosquito infection and viral replication). This sentence should be deleted, and Figure 2 should be amended by deleting the 0 dpi and the whole mosquito category.

- This sentence has been removed and Fig 2 updated as recommended

Lines 229-228: to be amended (see previous comment).

- Amended as recommended

Line 240-246: same comment on percentages to be rounded.

- Amended as recommended 

Line 246-247: I think the sentence is incomplete.

- This sentence has been altered 

Line 248-256: If the statistical tests are not significant, it means that the H0 hypothesis (no difference) cannot be rejected in view of your results. From there, why talk about an increase in viral titre? This section should be moderated in my opinion.

- There was a significant difference in the viral titre observed in the legs of mosquitoes at 7dpi however there were no other significant differences in viral titre in other body parts. This sentence has been altered to make this clear (line 263)

Figure 3: As for figure 2, representing the 0 dpi does not make sense. It is not a variable measured in the experiment, but an initial condition of the manipulation. In my opinion, 0 dpi should be removed from the figure.

- Figure and figure legend amended as recommended 

Line 297 - 298: ITs are not a measure of vector competence (i.e., EIP), but only of the susceptibility of mosquitoes to infection.

- This has been clarified throughout (see comments above)

Line 309 - 310: Yes, I agree with you on this point.

---

## [Decision Letter · Decision Letter 1]

10 Jul 2023

Knockdown resistance allele L1014F introduced by CRISPR/Cas9 is not associated with altered vector competence of Anopheles gambiae for o’nyong nyong virus

PONE-D-23-03395R1

Dear Dr. Reimer,

We’re pleased to inform you that your manuscript has been judged scientifically suitable for publication and will be formally accepted for publication once it meets all outstanding technical requirements.

Kind regards,

Pierre Roques, Ph.D.

Academic Editor

PLOS ONE

Additional Editor Comments (optional):

Reviewers' comments:

Reviewer's Responses to Questions

**Comments to the Author**

1. If the authors have adequately addressed your comments raised in a previous round of review and you feel that this manuscript is now acceptable for publication, you may indicate that here to bypass the “Comments to the Author” section, enter your conflict of interest statement in the “Confidential to Editor” section, and submit your "Accept" recommendation.

Reviewer #2: All comments have been addressed

2. Is the manuscript technically sound, and do the data support the conclusions?

Reviewer #2: Yes

3. Has the statistical analysis been performed appropriately and rigorously? 

Reviewer #2: Yes

4. Have the authors made all data underlying the findings in their manuscript fully available?

Reviewer #2: Yes

5. Is the manuscript presented in an intelligible fashion and written in standard English?

Reviewer #2: Yes

6. Review Comments to the Author

Reviewer #2: The revised version of the manuscript has addressed all concerned raised by previous reviewers. The revised version has improved the manuscript.

7. PLOS authors have the option to publish the peer review history of their article (what does this mean?). If published, this will include your full peer review and any attached files.

Reviewer #2: No

---

## [Editor Report · Acceptance letter]

2 Aug 2023

PONE-D-23-03395R1 

Knockdown resistance allele L1014F introduced by CRISPR/Cas9 is not associated with altered vector competence of *Anopheles gambiae* for o’nyong nyong virus 

Dear Dr. Reimer:

I'm pleased to inform you that your manuscript has been deemed suitable for publication in PLOS ONE. Congratulations! Your manuscript is now with our production department. 

Kind regards, 

on behalf of

Dr. Pierre Roques 

Academic Editor

PLOS ONE